# Hip, knee, and ankle joint forces during exoskeletal-assisted walking: Comparison of approaches to simulate human-robot interactions

**Gabriela B. De Carvalho**[1], **Vishnu D. Chandran**[2], **Ann M. Spungen**[3,4], **Noam Y. Harel**[ID][3,5], **William A. Bauman**[4], **Saikat Pal**[ID][1,3,6]*

1 Department of Biomedical Engineering, New Jersey Institute of Technology, Newark, New Jersey, United States of America, 2 Department of Rehabilitation, Hospital for Special Surgery, New York, New York, United States of America, 3 Spinal Cord Damage Research Center, James J. Peters Veterans Affairs Medical Center, Bronx, New York, United States of America, 4 Department of Medicine and Rehabilitation & Human Performance, Icahn School of Medicine at Mount Sinai, New York, New York, United States of America, 5 Department of Neurology, Icahn School of Medicine at Mount Sinai, New York, New York, United States of America, 6 Department of Electrical and Computer Engineering, New Jersey Institute of Technology, Newark, New Jersey, United States of America

* pal@njit.edu

## Abstract

The overall goal of this study was to develop a computational framework to quantify hip, knee, and ankle joint forces during exoskeletal-assisted walking (EAW) in the ReWalk P6.0, an FDA-approved lower-extremity exoskeleton. The first objective was to quantify hip, knee, and ankle joint forces during unassisted walking and compare the results to existing *in vivo* and simulation data. The second objective was to compute hip, knee, and ankle joint forces from four different approaches to simulate human-robot interactions during EAW. We recorded the three-dimensional motion of one able-bodied participant during unassisted walking and EAW, with simultaneous measurements of (i) marker trajectories, (ii) ground reaction forces, (iii) electromyography, and (iv) exoskeleton encoder data. We developed a subject-specific virtual simulator in OpenSim to reproduce unassisted walking and EAW. Next, we utilized OpenSim's extension, OpenSim Moco, to determine the joint reaction forces at the hips, knees, and ankles during unassisted walking and EAW. The computed peak hip, knee, and ankle joint compressive forces during unassisted walking were 3.42–3.82 body weight (BW), 3.10–3.48 BW, and 4.97–5.83 BW, respectively; these joint forces were comparable to prior *in vivo* and simulation results. The four approaches to simulate human-robot interactions during EAW resulted in peak compressive forces ranging from 2.98–4.66 BW, 2.82–5.83 BW, and 3.39–3.79 BW at the hip, knee, and ankle joints, respectively. This computational framework provides a low-risk and cost-effective technique to quantify the loading of the long bones and assess fracture risk during EAW in patients with severe bone loss in the lower extremities.

**Data availability statement:** Yes - all data are fully available without restriction; All relevant data are available on Figshare: https://doi.org/10.6084/m9.figshare.29891924.

**Funding:** Grant number: Department of Veterans Affairs under Grant VA RR&D # 1 I01 RX003561-01A2 Initials of authors: NYH, WAB, SP Full names of commercial companies that funded the study or authors: Not applicable Initials of authors who received salary or other funding from commercial companies: Not applicable URLs to sponsors' websites: https://www.research.va.gov/isrm/rrdt/ Grant number: National Science Foundation Graduate Research Fellowship Program under application #1000346414 (Fellow ID: 2022346414) Initials of authors: GBDC Full names of commercial companies that funded the study or authors: Not applicable Initials of authors who received salary or other funding from commercial companies: Not applicable URLs to sponsors' websites: https://www.nsfgrfp.org/ The sponsors or funders played no role in the study design, data collection and analysis, decision to publish, or preparation of the manuscript.

**Competing interests:** The authors have declared that no competing interests exist.

## Introduction

Approximately 308,000 persons live with spinal cord injury (SCI) in the United States, with around 18,000 new cases registered each year [1]. Major symptoms associated with SCI include partial or complete loss of sensory and motor function below the level of neurological lesion [2]. As a result, persons with SCI become wheelchair reliant for in-home and community ambulation. The loss of upright mobility following injury results in secondary symptoms, including chronic pain, muscle spasticity, bowel/bladder dysfunction, depression, and even premature death [3]. There is a clear need to restore upright mobility in persons with SCI to address these secondary symptoms and improve their quality of life.

Currently, the only clinical option to restore independent, upright mobility in persons with SCI is using wearable robotic exoskeletons. There are many benefits of upright over wheelchair ambulation, including access to wheelchair-inaccessible environments, psychological parity with able-bodied peers due to the ability to stand, improvements in cardiovascular and pulmonary function [4,5], and decreases in chronic pain [6–9], spasticity [6,7], and bone loss [10–14]. Paediatric exoskeletons have also been developed for different applications [15,16]. Prior studies have demonstrated the ability of persons with motor-incomplete and motor-complete SCI to ambulate independently in wearable robotic exoskeletons [5,17–30], highlighting the potential of these devices in restoring upright mobility. There are already four wearable robotic exoskeletons approved by the Food and Drug Administration (FDA) for persons with SCI, namely the ReWalk (ReWalk Robotics, Yoknaem, Israel), Ekso (Ekso Bionics, Richmond, CA), Ekso Indego (Ekso Bionics, Richmond, CA), and Atalante X (Wandercraft, New York City, NY). In January 2024, the Centers for Medicare and Medicaid Services approved reimbursement for the ReWalk device. It is only a matter of time before private insurance companies approve wearable robotic exoskeletons for ambulation as a component of standard care, highlighting the growing demand for these assistive devices in improving mobility and community integration for persons with SCI.

The growing demand for wearable robotic exoskeletons has given rise to a new clinical problem in persons with SCI and their clinicians, which is bone fractures during exoskeletal-assisted walking (EAW). Several studies have reported bone fractures in persons with SCI during exoskeletal-assisted movements [18,31–35], with an incidence rate of up to 10% [18,32]. An important secondary consequence of immobility after SCI is the rapid loss of bone strength below the neurological lesion [12–14]. Individuals with SCI lose a substantial amount of bone during the initial 12 months post injury [36], with the resorption of up to 73% of bone within the first few years [37,38]. During EAW, substantial forces are brought to bear on the lower limbs of users due to upright, weight-bearing locomotion. As a result, EAW places an already vulnerable SCI population at an increased risk of fracture.

Quantifying joint forces during EAW is an important step in assessing fracture risk in persons with SCI. However, knowledge of joint forces during EAW is currently limited. Experimental or *in vivo* methods to quantify joint forces during EAW, such as instrumented implants, are invasive. Although prior studies have reported *in vivo* joint

forces using instrumented implants during activities of daily living [39–44], recreating such experiments to quantify joint forces during EAW is practically infeasible. As such, computational simulation is the only viable alternative to quantify joint forces during EAW. Three prior simulation studies have reported knee joint forces during EAW [45–47]. McLain et al. used an EMG-informed neuromusculoskeletal model to estimate tibiofemoral contact forces from able-bodied persons walking with a custom knee exoskeleton [45]. Zhang et al. used a musculoskeletal model with generic gait data to estimate knee joint contact forces from able-bodied persons walking with a custom knee exoskeleton [46]. Dai et al. used a musculoskeletal model to simulate a gait intervention strategy to reduce knee joint loads with a rigid-soft hybrid exoskeleton [47]. However, no prior study has quantified joint forces during EAW in an FDA-approved exoskeleton. FDA approval is important because patients only have access to FDA-approved devices in clinics and for in-home use. Furthermore, no prior study has quantified hip or ankle joint forces during EAW in any exoskeleton.

The overall goal of this study was to develop a computational framework to quantify the hip, knee, and ankle joint forces during EAW in the ReWalk P6.0, an FDA-approved lower extremity exoskeleton. In pursuit of this overall goal and to gain confidence in our computational framework, we addressed the following research objectives on an able-bodied participant. The first objective of this study was to quantify hip, knee, and ankle joint forces during unassisted walking and compare the results to existing *in vivo* and simulation data; this direct comparison is only possible with able-bodied participants. The second objective of this study was to compute hip, knee, and ankle joint forces from four different approaches to simulate human-robot interactions during EAW in the ReWalk device. The four different approaches provide a relatively complete representation of the dynamics of EAW.

## Methods

### Participant recruitment

We recruited one able-bodied participant for this study (Fig 1A; male, age: 41 years, height: 1.76 m, mass: 89.4 kg). The Institutional Review Board at New Jersey Institute of Technology approved this study. The approval number is 2008001531R004. Written consent was obtained. The start date for recruitment for this study was October 1, 2021. The end date for recruitment for this study will be September 30, 2025.

### Exoskeletal-assisted walking training

The participant was trained to walk independently in the ReWalk P6.0 (Figs 1B and 1C). Prior to donning the exoskeleton, the device's physical dimensions and software configurations were adjusted to fit the participant's anthropometry and preferences, respectively. The training protocol followed standard procedures for walking in the ReWalk P6.0, and was described in detail in a prior publication [48]. Briefly, the participant was trained to shift their weight from side-to-side in an alternating manner to unload their contralateral leg, thereby triggering the activation of the hip and knee motors of the exoskeleton to initiate stepping. The participant was able to perform EAW in the exoskeleton independent of any external assistance after four one-hour sessions.

### Motion capture experiments during unassisted walking and EAW

We analyzed the three-dimensional (3-D) motion of the participant during unassisted walking and EAW from a single motion capture session (Figs 1D and 2A). The experimental protocol was described in detail in the prior publication [48]. Briefly, 3-D motion data included simultaneous measurements of marker trajectories, ground reaction forces, electromyography (EMG), and exoskeleton encoder data. Retro-reflective markers were tracked at 100 Hz using a 16-camera motion capture system (Vicon V8 and Nexus, Vicon Motion Systems, Oxford, UK; Fig 2B). Next, ground reaction forces were recorded from each foot at 2000 Hz using overground force plates (Bertec Corp., Columbus, OH; Fig 2C). Using established protocols, EMG measurements were recorded at 2000 Hz from rectus femoris, vastus lateralis, vastus medialis, semitendinosus, biceps femoris, gastrocnemius medialis, soleus, and tibialis anterior muscles with a 16-channel surface

## 3-D Motion Analysis to Subject-Specific Virtual Simulator of EAW

**Fig 1. A computational framework for quantifying hip, knee, and ankle joint forces during unassisted walking and EAW.** An able-bodied participant was recruited to perform EAW in the ReWalk P6.0 and their anthropometric measurements were collected (A). The physical dimensions and software configurations of the ReWalk P6.0 were adjusted to the participant's anthropometry and preferences, respectively (B). Next, the participant was trained to walk independently in the ReWalk P6.0 prior to a single session of motion capture experiments (C). Retro-reflective markers were placed on the participant (black and red), exoskeleton (blue), and hand crutches (green) to track the position and orientation of the human-robot system (D). The black markers represent the Conventional Gait Model (CGM) 2.5 marker template. The red markers are offset markers used to reconstruct occluded markers from the CGM 2.5 marker template when the participant is in the exoskeleton. The blue and green markers represent a custom marker template used to define the segments of the exoskeleton and hand crutches (D). The 3-D motion capture data were used as inputs to a subject-specific virtual simulator (E). The virtual simulator reproduced unassisted walking (F) and EAW (G-J) using EMG-tracked muscle driven simulations. The four approaches to simulate human-robot interactions during EAW were: 1) No Interactions (reproducing EAW kinematics but excluding exoskeletal interaction forces and motor torques; G); 2) Prescribed Torques (reproducing EAW kinematics and including exoskeletal motor torques prescribed directly at each corresponding human joint; H); 3) Bushing Forces (reproducing EAW kinematics and including interaction forces applied at the locations of the straps and the pelvic band, which are the points of contact between the human and the exoskeleton; I); and 4) a combination of Prescribed Torques and Bushing Forces (functionally the most similar to physical EAW; J).

EMG system (TrignoTM, Delsys Inc., Natick, MA; Fig 2D) [49,50]. These eight muscles are the large lower extremity muscles accessible through surface EMG. Exoskeleton encoder data were recorded during all trials of EAW, which included timestamps corresponding to motor activation and motor encoder angles (Fig 2E). The 3-D marker trajectories, ground reaction forces, EMG, and exoskeleton encoder data were synchronized.

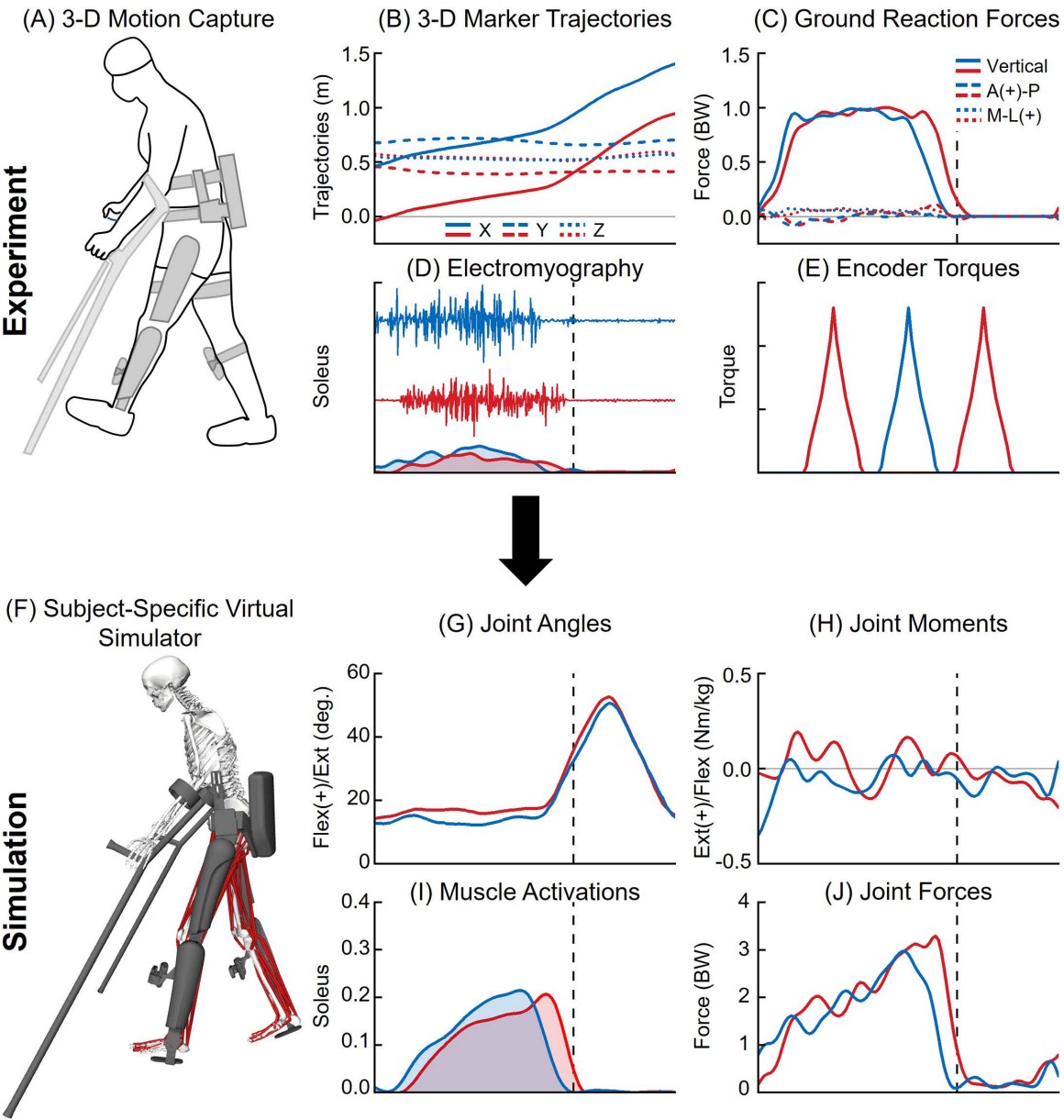

**Fig 2. 3-D motion capture experiment procedures and corresponding musculoskeletal simulations.** An able-bodied participant performed unassisted walking and EAW during a single motion capture session (A) to record 3-D motion data, including simultaneous measurements of marker trajectories (B), ground reaction forces (C), electromyography (EMG; D), and representative exoskeleton encoder data (E). The 3-D motion capture data (right leg: red lines, left leg: blue lines) were used as inputs to develop a subject-specific virtual simulator (F). Inverse kinematics and inverse dynamics analyses were performed to compute joint angles (G) and joint moments (H) during unassisted walking and EAW, respectively. Muscle activations (I) and joint forces (J) were computed using EMG-tracked muscle driven simulations of unassisted walking and EAW.

Three-dimensional motion data were collected while the participant performed unassisted walking and EAW on an instrumented walkway. The participant performed 10 unassisted walking trials at their self-selected speed (1.12±0.05 m/s) and 10 EAW trials at their preferred speed (0.47±0.03 m/s). Only successful trials were included for further analysis. A trial was considered successful if each foot struck only one force plate, and there were no missing ground reaction force

or EMG data. In addition, EAW trials were considered successful if hand crutches were positioned away from the force plates to avoid interference with ground reaction force data, and there were no missing exoskeleton encoder data. Based on these criteria, we obtained five successful trials of unassisted walking and six successful trials of EAW.

## Virtual simulator reproducing unassisted walking and EAW

Unassisted walking and EAW were reproduced using a virtual simulator developed in OpenSim (Figs 1E and 2F) [51]. The framework to develop the subject-specific virtual simulator was described in detail in the prior publication [48]. Briefly, a generic human musculoskeletal model was scaled to match the participant's body mass and segment dimensions using 3-D marker trajectories from a static trial [52]. A full-scale geometry of the ReWalk P6.0 exoskeleton was integrated with the scaled human model.

We previously published the dynamics (joint angles and joint torques) of unassisted walking and EAW from this dataset [48]. Briefly, we quantified joint angles using inverse kinematics analyses in OpenSim, which minimizes the errors between experimental markers and the corresponding virtual markers on the human-robot model (Fig 2G) [53,54]. Next, we quantified net human-robot joint torques using inverse dynamics analyses in OpenSim (Fig 2H) [53,54]. For inverse dynamics analyses, we combined the mass and inertial properties of exoskeleton segments with corresponding human segments to simplify the human-robot model, similar to previous studies [55,56]. Exoskeleton masses were added to the respective segments in the musculoskeletal model. In addition, the centers of mass of the human segments were moved to the locations corresponding to the centers of mass of the combined human-robot segments.

We utilized OpenSim's extension, OpenSim Moco [57], to determine the joint reaction forces of the hips, knees, and ankles during unassisted walking and EAW (Figs 1F-1J). We chose OpenSim Moco over static optimization or CMC due to its robustness in solving optimization problems [58] and incorporating muscle dynamics [57,59,60]. We adapted OpenSim Moco's *MocoInverse* tool to perform EMG-tracked muscle driven simulations of unassisted walking and EAW [57]. The *MocoInverse* tool solves for the muscle or actuator controls required to achieve a prescribed motion. We used two default cost terms from OpenSim Moco's *MocoInverse* tool: 1) minimizing the sum of squared muscle activations (Fig 2I), and 2) constrain initial activations to initial excitations for each muscle. In addition, we modified the cost function to include a third cost term, which is EMG tracking of the eight muscles for which experimental EMG data were collected. We replaced the model's Millard2012Equilibrium muscles [61] with the DeGrooteFregly2016 muscles [58] to make the model well-suited for optimal control problems in OpenSim Moco [57]. Muscle control values were constrained from 0 to 1. Muscle activation dynamics and tendon compliance were maintained in the model. Tendon compliance dynamics were set to "implicit" mode, which is favored in optimization problems due to its robustness and computational efficiency [58]. Passive muscle fiber forces were assumed to be zero, which is reasonable for simulating walking. The locked subtalar and metatarsophalangeal joints were replaced with welded joints in our simulations. We applied the left and right ground reaction forces to each respective calcaneus in the model. For EAW trials, we subtracted the weight of the exoskeleton (294 N) from the ground reaction force data to account for only the body weight of the human. We prescribed the kinematics of the model using the joint angles from the inverse kinematics analyses for each unassisted walking and EAW trial. We provided the experimental EMG data as the tracking reference for the eight lower limb muscles on each leg. Joint reaction forces at the hips, knees, and ankles were computed in the ground reference frame using the output states and controls from the *MocoInverse* solution (Fig 2J) [57]. Joint reaction forces comprised the contributions of external forces and forces from all muscles crossing that joint. Joint reaction forces were transformed into the reference frames of the child bodies for each joint using a custom script in Matlab.

## Four approaches to simulate human-robot interactions during EAW

We simulated EAW using four different methods of increasing complexity to gain confidence in our computational framework (Figs 1G-1J).

**Method 1: EAW (No Interactions).** The virtual simulator reproduced the kinematics of the human during EAW, but excluded contributions from exoskeletal motor torques and interaction forces (Fig 1G).

**Method 2: EAW (Prescribed Torques).** The virtual simulator reproduced the kinematics of the human during EAW and included contributions from exoskeletal motor torques prescribed directly at each corresponding human joint (Fig 1H). Exoskeletal motor torques were defined in OpenSim's ground reference frame. Equal and opposite body torques were applied to the tibia and the femur to define the knee motor's flexion-extension torque. Similarly, equal and opposite body torques were applied to the femur and the pelvis to define the hip motor's flexion-extension torque. The directions of exoskeleton motor torques were superimposed onto the flexion-extension axes of the human joints in the musculoskeletal model.

**Method 3: EAW (Bushing Forces).** The virtual simulator reproduced the kinematics of the human during EAW and included contributions from exoskeletal interaction forces applied at locations of the straps and the pelvic band, which are the points of contact between the human and the exoskeleton (Fig 1I). The interaction forces were distributed across body segments consistent with the locations of the straps and pelvic band. These locations were obtained from experimental data. The distribution of bushing elements in the musculoskeletal model was as follows: one at the torso, two at each thigh, and one at each shank (Fig 1I). To estimate human-robot interaction forces, we applied OpenSim's bushing force element, a 3-D linear spring-damper system, to a musculoskeletal model including the exoskeleton geometry. Translational stiffness and damping parameters were tuned using experimental data from an instrumented exoskeleton knee bracket. Rotational parameters were assumed to be 0 (default in OpenSim). Translational and rotational bushing force parameters were assumed to be the same at all exoskeletal straps/bands. Bushing forces and their corresponding torques and points of application were estimated using OpenSim's ForceReporter and PointKinematics analyses for the entire duration of each EAW trial. Next, these results were included in an external force file together with the ground reaction forces for the respective trial to perform the EMG-tracked muscle driven simulations of EAW on a musculoskeletal model excluding the exoskeleton.

**Method 4: EAW (Prescribed Torques + Bushing Forces).** The virtual simulator reproduced the kinematics of the human during EAW and included contributions from both exoskeletal motor torques and interaction forces (Fig 1J). This approach was functionally the most similar to actual physical EAW.

## Data analysis and statistical methods

We quantified compressive, anterior-posterior (A-P), and medial-lateral (M-L) hip, knee, and ankle joint forces from the five trials of unassisted walking and six trials of EAW. For each trial, the joint forces were normalized to the participant's body weight. The joint forces from multiple trials were averaged for each leg during unassisted walking and the four approaches to simulate human-robot interactions during EAW. We compared joint forces from unassisted walking to previously published *in vivo* and simulation data. Next, we compared joint forces from the four approaches to simulate human-robot interactions during EAW.

We compared hip, knee, and ankle flexion-extension angles from our EMG-tracked muscle driven simulations to inverse kinematics data from the five trials of unassisted walking and six trials of EAW. The joint angles from multiple trials were averaged for each leg during unassisted walking and the four approaches to simulate human-robot interactions during EAW. For each trial, root mean square (RMS) errors between the joint angles from EMG-tracked muscle driven simulations and corresponding inverse kinematics data were calculated. The RMS error values from multiple trials were averaged.

We compared hip, knee, and ankle flexion-extension moments from our EMG-tracked muscle driven simulations to inverse dynamics data from the five trials of unassisted walking and six trials of EAW. The computed joint moments from the EMG-tracked muscle driven simulations included contributions that differed between the approaches as follows: 1) unassisted walking and EAW (No Interactions) included contributions from only muscle forces; 2) EAW (Prescribed

Torques) included contributions from muscle forces and exoskeletal motor torques; 3) EAW (Bushing Forces) included contributions from muscle forces and interaction forces applied at the locations of the straps and the pelvic band; and 4) EAW (Prescribed Torques + Bushing Forces) included contributions from muscle forces, exoskeletal motor torques, and interaction forces applied at the locations of the straps and the pelvic band. For the EMG-tracked muscle driven simulations, joint moments from both unassisted walking and EAW trials were normalized to the participant's body mass. For the inverse dynamics trials, joint moments from unassisted walking were normalized to the participant's body mass, and joint moments from EAW were normalized to the combined mass of the participant and the exoskeleton. The joint moments from multiple trials were averaged for each leg during unassisted walking and the four approaches to simulate human-robot interactions during EAW. For each trial, RMS errors between the joint moments from EMG-tracked muscle driven simulations and corresponding inverse dynamics data were calculated. The RMS error values from multiple trials were averaged.

We compared muscle activations from our EMG-tracked muscle driven simulations to normalized experimental EMG data from the five trials of unassisted walking and six trials of EAW. The simulated muscle activations and normalized EMG data from multiple trials were averaged for each leg during unassisted walking and the four approaches to simulate human-robot interactions during EAW.

Appropriate statistical methods add scientific rigor and help with data interpretation. Our study includes a single able-bodied participant. From this one participant, we recorded multiple trials of unassisted walking and EAW, which we presented with appropriate average and standard deviation results. Comparison of unassisted walking vs. EAW through statistical tests such as t-tests or ANOVA was not a research objective of this study. Next, we developed virtual simulators of unassisted walking and compared our computed joint forces to prior *in vivo* and simulation studies. Here, we appropriately reported RMS errors; additional comparisons through t-tests or ANOVA with missing data from prior published studies is not statistically sound. Finally, we developed virtual simulators from four different approaches to simulate human-robot interactions during EAW to test the robustness of our computational framework. Comparison of the results from the four different simulation conditions through statistical tests such as t-tests or ANOVA was not a research objective of this study.

## Results

The virtual simulator closely reproduced unassisted walking and EAW, with average (±1 SD) RMS errors between experimental and simulator markers being 1.19 (± 0.02) cm and 1.27 (± 0.06) cm for unassisted walking and EAW trials, respectively. Per OpenSim guidelines, the acceptable tolerance between experimental and simulator markers is an average RMS error of less than 2.00 cm [62,63].

Our computed hip, knee, and ankle compressive joint forces during unassisted walking were comparable to previously published *in vivo* and simulation studies (Fig 3). For the hip joint, peak computed compressive forces averaged over five trials of unassisted walking were between 3.42 body weight (BW) and 3.82 BW; in comparison, peak compressive forces from prior *in vivo* and simulation studies ranged from 2.31 BW to 4.33 BW [39,64] (Figs 3A and 3B). Average (±1 SD) RMS errors between computed compressive forces and *in vivo* results were 0.91 (± 0.07) BW (left, Fig 3A) and 0.71 (± 0.10) BW (right, Fig 3B), and previously published simulation results were 0.63 (± 0.06) BW (left, Fig 3A) and 0.56 (± 0.06) BW (right, Fig 3B). For the knee joint, peak computed compressive forces averaged over five trials of unassisted walking were between 3.10 BW and 3.48 BW; in comparison, peak compressive forces from prior *in vivo* studies ranged from 2.32 BW to 2.36 BW [42,44] (Figs 3C and 3D). Average (±1 SD) RMS errors between computed compressive forces and *in vivo* results from Fregly et al. [42] were 0.55 (± 0.04) BW (left, Fig 3C) and 0.57 (± 0.04) BW (right, Fig 3D), and *in vivo* results from Kutzner et al. [44] were 0.91 (± 0.05) BW (left, Fig 3C) and 0.90 (± 0.04) BW (right, Fig 3D). For the ankle joint, peak computed compressive forces averaged over five trials of unassisted walking were between 4.97 BW and 5.83 BW; in comparison, peak compressive force from a prior simulation study was 5.58 BW [65] (Figs 3E and 3F). Average

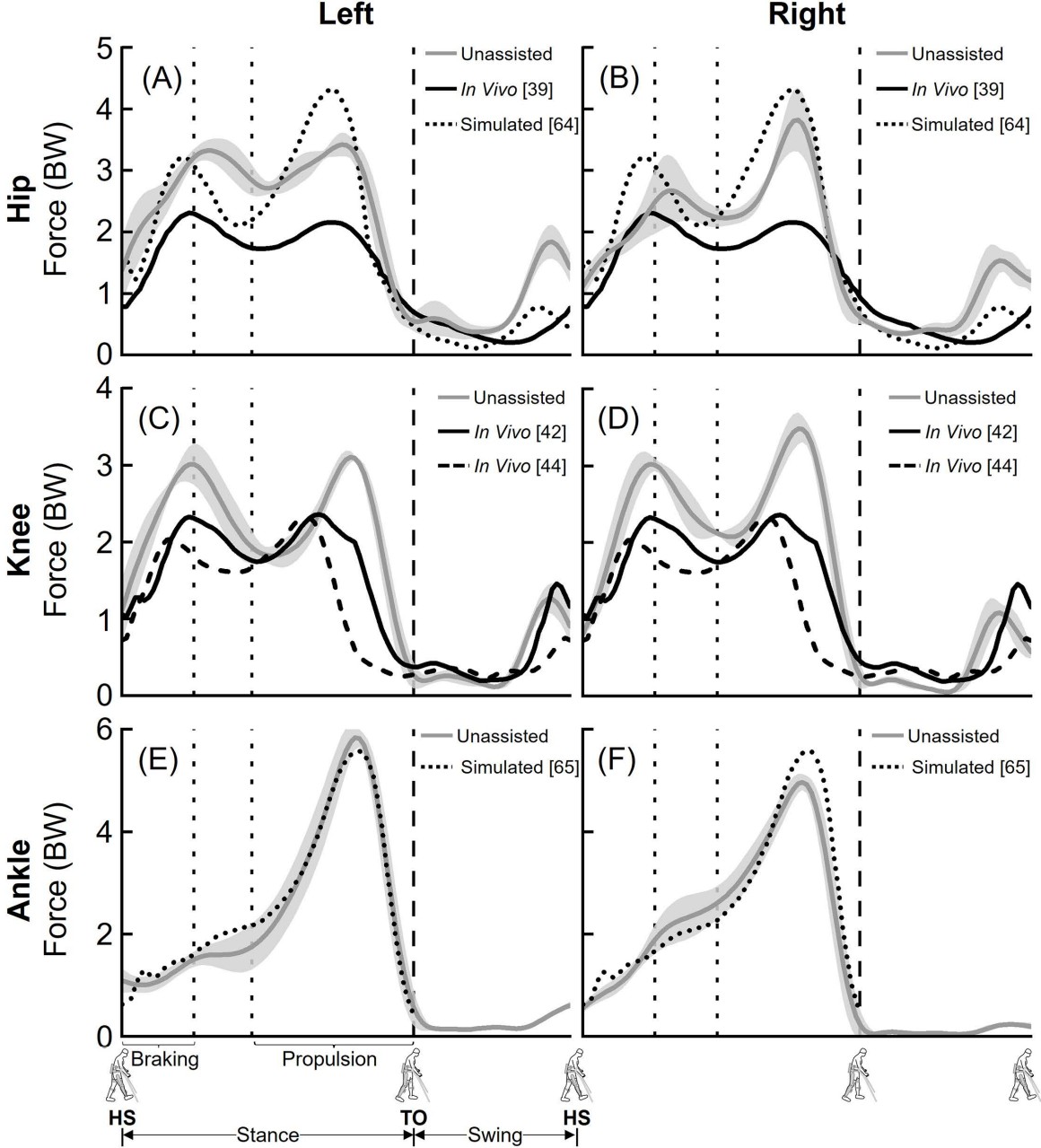

**Fig 3. Average (±1 SD) hip (A-B), knee (C-D), and ankle (E-F) compressive joint forces during unassisted walking (five trials, grey lines).** Hip compressive forces were compared to prior *in vivo* (solid black lines) [39] and simulation (dotted black lines) [64] studies (A-B). Knee compressive forces were compared to two prior *in vivo* studies (solid and dashed black lines) [42,44] (C-D). Ankle compressive forces were compared to a prior simulation study (dotted black lines) [65] (E-F). The joint forces were normalized to the participant's body weight (BW). Average toe-off from all unassisted walking trials is represented by dashed vertical lines. The braking and propulsion phases of gait during unassisted walking are represented by dotted vertical lines.

(±1 SD) RMS errors between computed compressive forces and previously published simulation results were 0.47 (± 0.18) BW (left, Fig 3E) and 0.55 (± 0.14) BW (right, Fig 3F). Comparisons of computed anterior-posterior and medial-lateral hip, knee, and ankle joint forces during unassisted walking are provided as supplemental information (S1-S4 Figs).

The four approaches to simulate human-robot interactions during EAW resulted in a range of peak compressive forces at the hip and knee joints (Figs 4 and 5). For the hip joint, peak computed compressive forces averaged over six trials of EAW were 3.10–3.41 BW, 2.98–3.68 BW, 3.76–4.10 BW, and 4.08–4.66 BW for EAW (No Interactions), EAW (Prescribed Torques), EAW (Bushing Forces), and EAW (Prescribed Torques + Bushing Forces), respectively (Figs 4A-4D and 5A-5D). For the knee joint, peak computed compressive forces averaged over six trials of EAW were 2.82–2.84 BW, 3.50–4.95 BW, 3.54–3.55 BW, and 4.58–5.83 BW for EAW (No Interactions), EAW (Prescribed Torques), EAW (Bushing Forces), and EAW (Prescribed Torques + Bushing Forces), respectively (Figs 4E-4H and 5E–5H). For the ankle joint, the four approaches to simulate human-robot interactions during EAW resulted in similar peak compressive forces, ranging from 3.39–3.79 BW (Figs 4I-4L and 5I–5L). Comparisons of computed anterior-posterior and medial-lateral hip, knee, and ankle joint forces from the four approaches to simulate human-robot interactions during EAW are provided as supplemental information (S1-S4 Figs). It is important to point out that external crutch reaction forces were not recorded during our experiments, nor included in our virtual simulator. The handheld crutches permit additional stability and likely offload the lower limb joints during EAW. It is likely that the absence of crutch forces resulted in overestimation of joint loads during EAW.

The EMG-tracked muscle driven simulations of unassisted walking and EAW closely matched our joint angles from inverse kinematics (Figs 6 and 7). Average (±1 SD) RMS errors were within 0.4 (± 0.2)°, 0.8 (± 0.6)°, and 0.4 (± 0.2)° for the hip, knee, and ankle joints, respectively (Figs 6 and 7). Next, the EMG-tracked muscle driven simulations of unassisted walking and EAW closely matched our joint moments from inverse dynamics (Figs 8 and 9). For unassisted

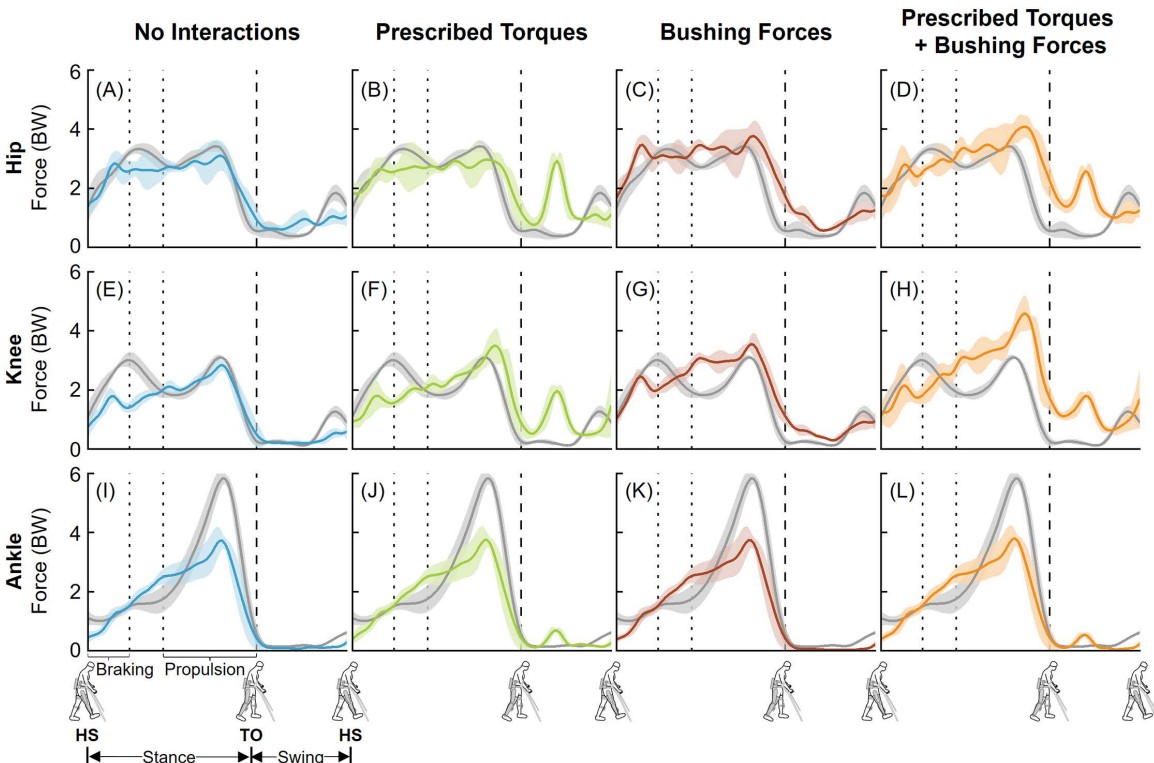

**Fig 4. Left leg compressive joint forces from the four approaches to simulate human-robot interactions during EAW.** Average (±1 SD) hip (A-D), knee (E-H), and ankle (I-L) joint forces during EAW (six trials, colored lines) were compared to unassisted walking (five trials, grey lines). The joint forces were normalized to the participant's body weight (BW). Average toe-off from all unassisted walking and EAW trials is represented by dashed vertical lines. The braking and propulsion phases of gait during unassisted walking are represented by dotted vertical lines.

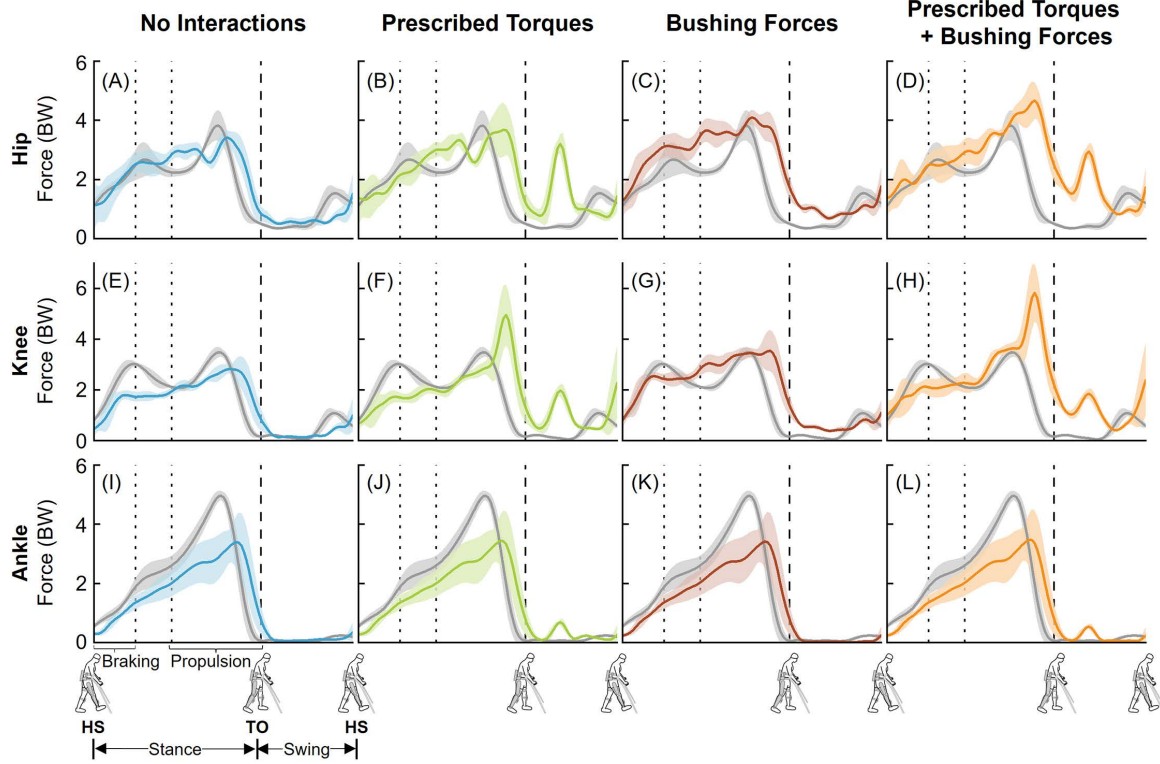

**Fig 5. Right leg compressive joint forces from the four approaches to simulate human-robot interactions during EAW.** Average (±1 SD) hip (A-D), knee (E-H), and ankle (I-L) joint forces during EAW (six trials, colored lines) were compared to unassisted walking (five trials, grey lines). The joint forces were normalized to the participant's body weight (BW). Average toe-off from all unassisted walking and EAW trials is represented by dashed vertical lines. The braking and propulsion phases of gait during unassisted walking are represented by dotted vertical lines.

walking, average (±1 SD) RMS errors between EMG-tracked muscle driven simulations (contributions from only muscle forces) and inverse dynamics were within 0.08 (± 0.01) Nm/kg (Figs 8A, 8F, 8K, 9A, 9F, and 9K). For EAW (No Interactions), average (±1 SD) RMS errors between EMG-tracked muscle driven simulations (contributions from only muscle forces) and inverse dynamics were within 0.06 (± 0.01) Nm/kg (Figs 8B, 8G, 8L, 9B, 9G, and 9L). For EAW (Prescribed Torques), average (±1 SD) RMS errors between EMG-tracked muscle driven simulations (contributions from muscle forces and exoskeletal motor torques) and inverse dynamics were within 0.06 (± 0.01) Nm/kg (Figs 8C, 8H, 8M, 9C, 9H, and 9M). For EAW (Bushing Forces), average (±1 SD) RMS errors between EMG-tracked muscle driven simulations (contributions from muscle forces and interaction forces applied at the locations of the straps and the pelvic band) and inverse dynamics were within 0.11 (± 0.01) Nm/kg (Figs 8D, 8I, 8N, 9D, 9I, and 9N). For EAW (Prescribed Torques + Bushing Forces), average (±1 SD) RMS errors between EMG-tracked muscle driven simulations (contributions from muscle forces, exoskeletal motor torques, and interaction forces applied at the locations of the straps and the pelvic band) and inverse dynamics were within 0.11 (± 0.01) Nm/kg (Figs 8E, 8J, 8O, 9E, 9J, and 9O). The small RMS errors in joint angles and joint moments demonstrate the ability of the virtual simulator to replicate the dynamics of unassisted walking and all four approaches to simulate human-robot interactions during EAW.

Next, our predicted muscle activations from the EMG-tracked muscle driven simulations were generally in agreement with the measured experimental EMG data (Figs 10 and 11). The greatest differences between the simulations and measured EMG data were for the soleus muscle (Figs 10AE-10AI and 11AE-11AI).

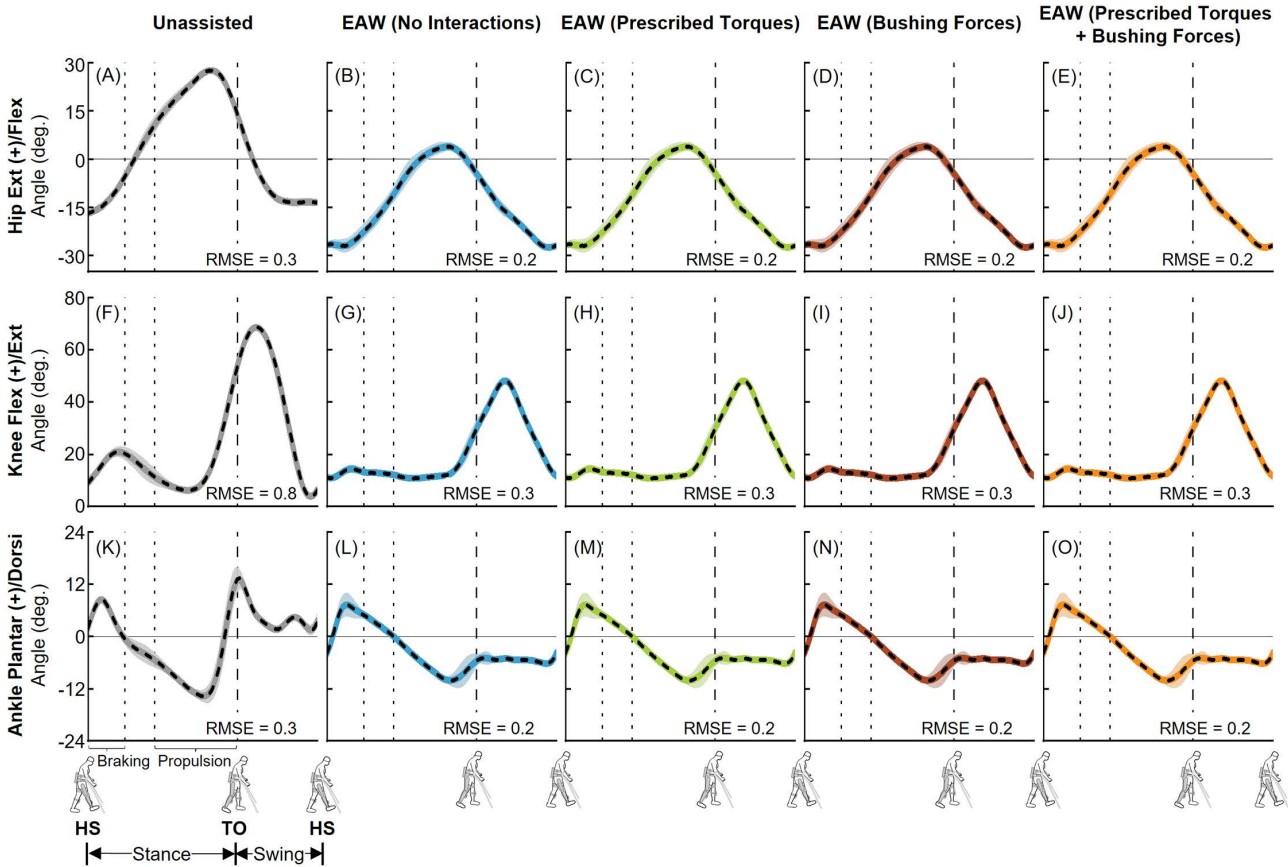

**Fig 6. Left leg joint kinematics from unassisted walking and the four approaches to simulate human-robot interactions during EAW.** Average (±1 SD) hip (A-E), knee (F-J), and ankle (K-O) joint angles from our EMG-tracked muscle driven simulations of unassisted walking (five trials, grey lines) and EAW (six trials, colored lines) were compared to inverse kinematics (IK, dashed black lines) data. The root mean square errors (RMSE) between EMG-tracked muscle driven simulations and IK were calculated for each trial and average RMSE for all trials are shown. Average toe-off from respective unassisted walking or EAW trials is represented by dashed vertical lines. The braking and propulsion phases of gait during unassisted walking are represented by dotted vertical lines.

## Discussion

The goal of this study was to develop a computational framework to quantify the hip, knee, and ankle joint forces during EAW in the ReWalk P6.0, an FDA-approved lower extremity exoskeleton. The first objective was to quantify hip, knee, and ankle joint forces during unassisted walking and compare the results to existing *in vivo* and simulation data. Our computed peak hip, knee, and ankle joint compressive forces during unassisted walking were 3.42–3.82 BW, 3.10–3.48 BW, and 4.97–5.83 BW, respectively (Fig 3). These results were comparable to published *in vivo* and simulation data (Fig 3). The second objective of this study was to compute hip, knee, and ankle joint forces from four different approaches to simulate human-robot interactions during EAW in the ReWalk device. The four approaches resulted in a range of peak compressive forces, with 2.98–4.66 BW at the hip joints, 2.82–5.83 BW at the knee joints, and 3.39–3.79 BW at the ankle joints (Figs 4 and 5).

We are not aware of any prior literature that has quantified hip, knee, and ankle joint forces during EAW in an FDA-approved lower extremity exoskeleton. Three prior simulation studies have reported knee joint forces during EAW [45–47]. McLain et al. used an EMG-informed neuromusculoskeletal model to report a reduction in tibiofemoral contact forces with

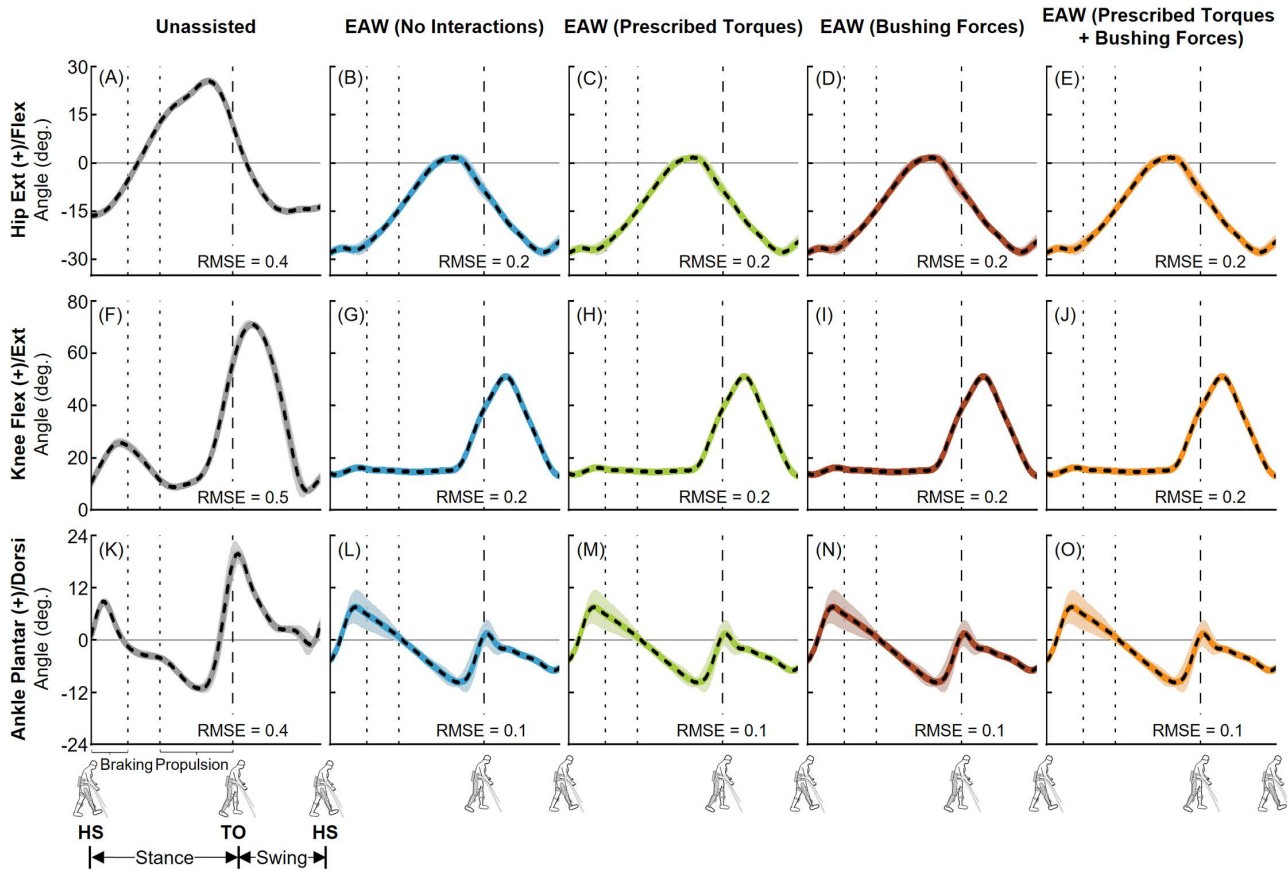

**Fig 7. Right leg joint kinematics from unassisted walking and the four approaches to simulate human-robot interactions during EAW.** Average (±1 SD) hip (A-E), knee (F-J), and ankle (K-O) joint angles from our EMG-tracked muscle driven simulations of unassisted walking (five trials, grey lines) and EAW (six trials, colored lines) were compared to inverse kinematics (IK, dashed black lines) data. The root mean square errors (RMSE) between EMG-tracked muscle driven simulations and IK were calculated for each trial and average RMSE for all trials are shown. Average toe-off from respective unassisted walking or EAW trials is represented by dashed vertical lines. The braking and propulsion phases of gait during unassisted walking are represented by dotted vertical lines.

exoskeletal knee extension assistance during the early stance phase of gait [45]. Zhang et al. used a musculoskeletal model to study the effects of different knee assistive strategies on knee contact forces [46]. Furthermore, Zhang et al. performed simulations using generic gait data from unassisted walking, under the assumption that joint kinematics from unassisted walking and EAW were the same. Next, Dai et al. reported that a gait intervention strategy reduced knee joint loads during EAW in a rigid-soft hybrid exoskeleton [47]. All three studies used custom-built exoskeletons that are not FDA-approved and thus, not readily available for the rehabilitation of persons with SCI. To the best of our knowledge, this is also the first study to report hip and ankle joint forces during EAW in any exoskeleton.

Our computational framework provides a logical approach to validate the computed joint forces during EAW in the absence of *in vivo* data. To establish confidence in our computational framework, we compared our joint forces during unassisted walking to previously published *in vivo* and simulation data (Fig 3). This direct comparison is only possible with able-bodied participants. Our computed hip joint compressive forces during unassisted walking were higher than published *in vivo* data [39], but comparable to results from a prior simulation study [64] (Fig 3). Next, our computed knee and ankle joint compressive forces were in close agreement with published *in vivo* [42,44] and simulation data [65], with

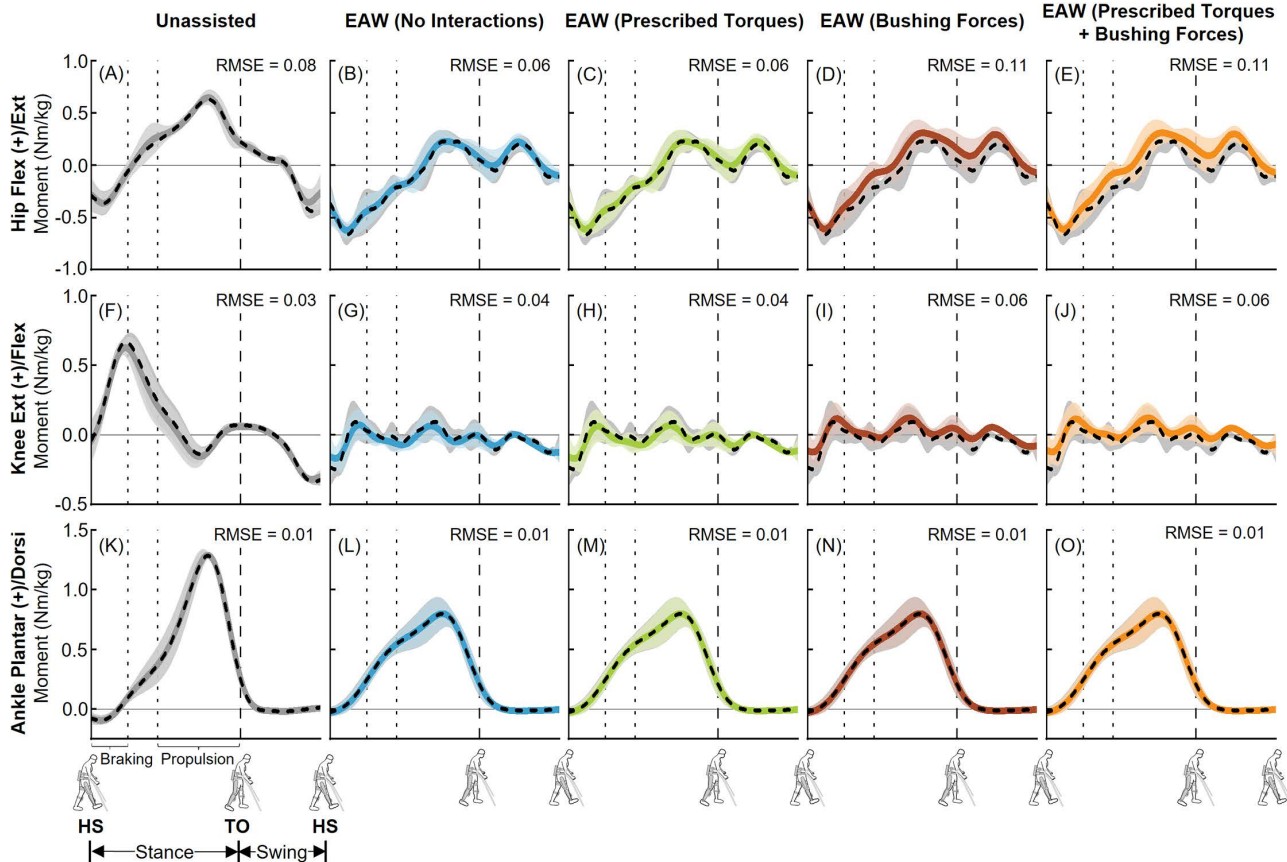

**Fig 8. Left leg joint moments from unassisted walking and the four approaches to simulate human-robot interactions during EAW.** Average (±1 SD) hip (A-E), knee (F-J), and ankle (K-O) joint moments from our EMG-tracked muscle driven simulations of unassisted walking (five trials, grey lines) and EAW (six trials, colored lines) were compared to inverse dynamics (ID, dashed black lines) data. The computed joint moments from unassisted walking (A, F, K) and EAW (No Interactions; B, G, L) included contributions from only muscle forces. The computed joint moments from EAW (Prescribed Torques) included contributions from muscle forces and exoskeletal motor torques (C, H, M). The computed joint moments from EAW (Bushing Forces) included contributions from muscle forces and interaction forces applied at the locations of the straps and the pelvic band (D, I, N). The computed joint moments from EAW (Prescribed Torques + Bushing Forces) included contributions from muscle forces, exoskeletal motor torques, and interaction forces applied at the locations of the straps and the pelvic band (E, J, O). For the EMG-tracked muscle driven simulations, joint moments from both unassisted walking and EAW trials were normalized to the participant's body mass. For the ID trials, joint moments from unassisted walking were normalized to the participant's body mass, and joint moments from EAW were normalized to the combined mass of the participant and the exoskeleton. The root mean square errors (RMSE) between EMG-tracked muscle driven simulations and ID were calculated for each trial and average RMSE for all trials are shown. Average toe-off from respective unassisted walking or EAW trials is represented by dashed vertical lines. The braking and propulsion phases of gait during unassisted walking are represented by dotted vertical lines.

average (±1 SD) RMS errors within 0.91 (± 0.05) BW and 0.55 (± 0.14) BW for the knee and ankle joints, respectively. These comparisons to prior studies provide confidence in our computational framework and set the foundation for computing joint forces during EAW, for which *in vivo* studies are not feasible.

Our virtual simulator reproduced EAW using four different simulation approaches of increasing complexity, resulting in a range of computed joint forces. Our first simulation approach, EAW (No Interactions), reproduced the kinematics of the human during EAW, but excluded contributions from exoskeletal motor torques and interaction forces (Fig 1G). This simulation approach required the same set of input parameters (joint kinematics, experimental EMG, and measured ground reaction forces) as unassisted walking. As a result, the computed joint forces from this simulation approach were similar

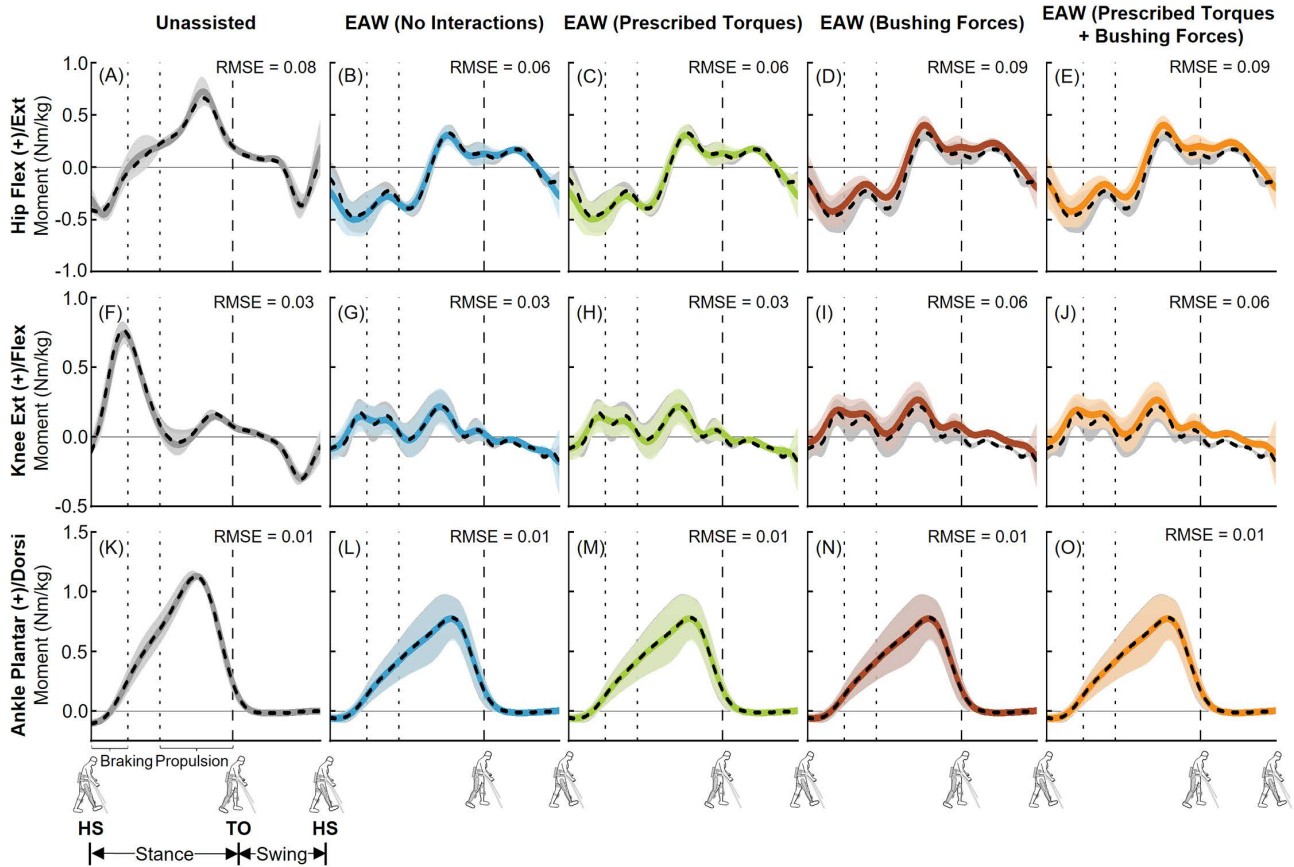

**Fig 9. Right leg joint moments from unassisted walking and the four approaches to simulate human-robot interactions during EAW.** Average (±1 SD) hip (A-E), knee (F-J), and ankle (K-O) joint moments from our EMG-tracked muscle driven simulations of unassisted walking (five trials, grey lines) and EAW (six trials, colored lines) were compared to inverse dynamics (ID, dashed black lines) data. The computed joint moments from unassisted walking (A, F, K) and EAW (No Interactions; B, G, L) included contributions from only muscle forces. The computed joint moments from EAW (Prescribed Torques) included contributions from muscle forces and exoskeletal motor torques (C, H, M). The computed joint moments from EAW (Bushing Forces) included contributions from muscle forces and interaction forces applied at the locations of the straps and the pelvic band (D, I, N). The computed joint moments from EAW (Prescribed Torques + Bushing Forces) included contributions from muscle forces, exoskeletal motor torques, and interaction forces applied at the locations of the straps and the pelvic band (E, J, O). For the EMG-tracked muscle driven simulations, joint moments from both unassisted walking and EAW trials were normalized to the participant's body mass. For the ID trials, joint moments from unassisted walking were normalized to the participant's body mass, and joint moments from EAW were normalized to the combined mass of the participant and the exoskeleton. The root mean square errors (RMSE) between EMG-tracked muscle driven simulations and ID were calculated for each trial and average RMSE for all trials are shown. Average toe-off from respective unassisted walking or EAW trials is represented by dashed vertical lines. The braking and propulsion phases of gait during unassisted walking are represented by dotted vertical lines.

to unassisted walking, especially at the hip (Figs 4A and 5A) and knee joints (Figs 4E and 5E). Our second simulation approach, EAW (Prescribed Torques), reproduced the kinematics of the human during EAW and included contributions from exoskeletal motor torques prescribed directly at each corresponding human joint (Fig 1H). Addition of these motor torques increased hip and knee joint compressive forces during the swing phase (Figs 4B, 4F, 5B, and 5F), when the robot's hip and knee motors were activated to facilitate directional changes in the robot segments (extension to flexion at the hip, flexion to extension at the knee) and prepare for heel strike. Our third simulation approach, EAW (Bushing Forces), reproduced the kinematics of the human during EAW and included contributions from exoskeletal interaction forces applied at locations of the straps and the pelvic band (Fig 1I). Addition of these bushing forces increased peak hip

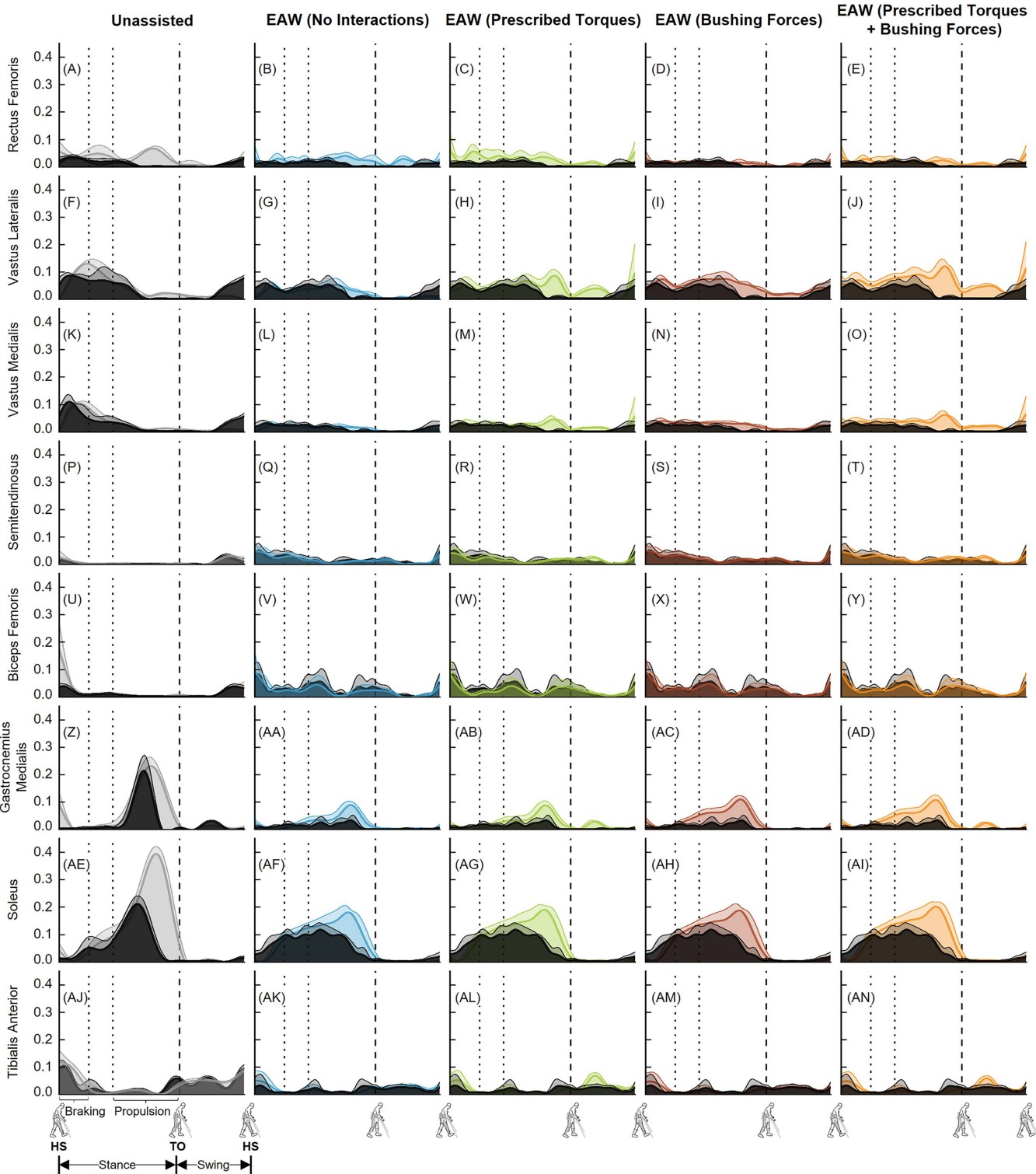

**Fig 10. Left leg muscle activations from unassisted walking and the four approaches to simulate human-robot interactions during EAW.** Average (+1 SD) computed muscle activations from EMG-tracked muscle driven simulations of unassisted walking (five trials, grey lines) and EAW (six trials, colored lines) were compared to experimental EMG measurements (black lines). The experimental EMG data were normalized using maximum muscle activations measured from maximum voluntary contraction trials. Average toe-off from respective unassisted walking or EAW trials is represented by dashed vertical lines. The braking and propulsion phases of gait during unassisted walking are represented by dotted vertical lines.

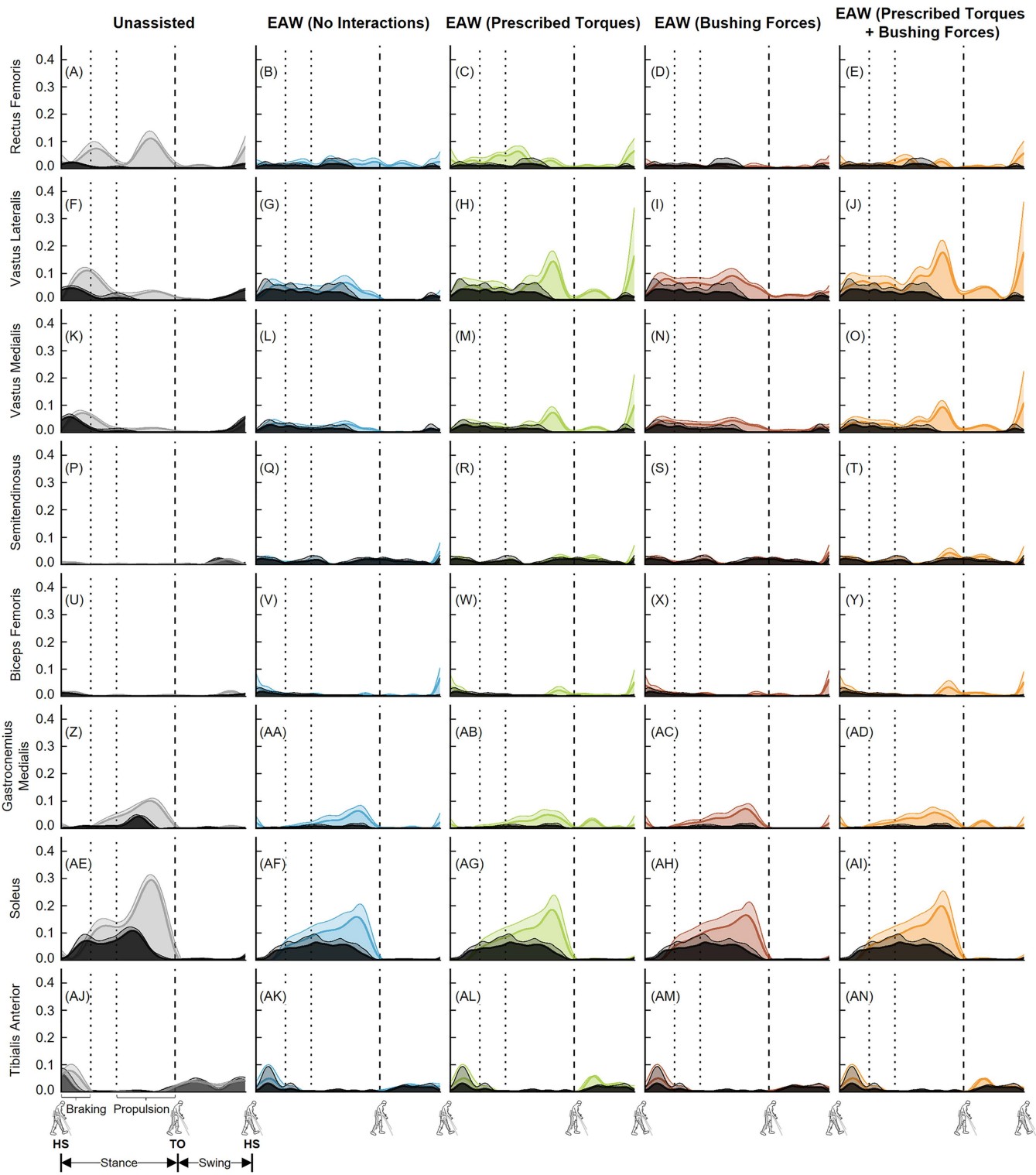

**Fig 11. Right leg muscle activations from unassisted walking and the four approaches to simulate human-robot interactions during EAW.**
Average (+1 SD) computed muscle activations from EMG-tracked muscle driven simulations of unassisted walking (five trials, grey lines) and EAW (six trials, colored lines) were compared to experimental EMG measurements (black lines). The experimental EMG data were normalized using maximum muscle activations measured from maximum voluntary contraction trials. Average toe-off from respective unassisted walking or EAW trials is represented by dashed vertical lines. The braking and propulsion phases of gait during unassisted walking are represented by dotted vertical lines.

and knee joint compressive forces by up to 21.2% and 25.7%, respectively, compared to EAW (No Interactions) (Figs 4C, 4G, 5C, and 5G). Our fourth simulation approach, EAW (Prescribed Torques + Bushing Forces), reproduced the kinematics of the human during EAW and included contributions from both exoskeletal motor torques and interaction forces (Fig 1J). This simulation approach was functionally the most similar to actual physical EAW, and the computed hip and knee joint forces are likely most representative of joint forces during EAW (Figs 4D, 4H, 5D, and 5H). Ankle joint compressive forces did not differ substantially between simulation approaches because exoskeletal contributions were only applied to the hip and knee joints (Figs 4I-4L and 5I–5L).

The virtual simulator was able to reproduce the dynamics of unassisted walking and all four approaches to simulate human-robot interactions during EAW. Joint angles from our EMG-tracked muscle driven simulations of unassisted walking and all four approaches to simulate human-robot interactions during EAW closely matched our inverse kinematics data, with average RMS errors up to 0.4° at the hips (Figs 6A-6E and 7A-7E), 0.8° at the knees (Figs 6F-6J and 7F–7J), and 0.4° at the ankles (Figs 6K-6O and 7K–7O). Similarly, joint moments from our EMG-tracked muscle driven simulations of unassisted walking and the four approaches to simulate human-robot interactions during EAW closely matched our inverse dynamics data, with average RMS errors up to 0.11 Nm/kg at the hips (Figs 8A-8E and 9A–9E), 0.06 Nm/kg at the knees (Figs 8F-8J and 9F–9J), and 0.01 Nm/kg at the ankles (Figs 8K-8O and 9K–9O). During EAW, the human-robot system represents complex dynamic interactions, and separating the joint moment contributions of the human from the robot is not well-understood. For unassisted walking and EAW (No Interactions) conditions, joint moments from the EMG-tracked muscle driven simulations comprised contributions from only muscle forces. For the EAW (Prescribed Torques) condition, joint moments from the EMG-tracked muscle driven simulations comprised contributions from muscle forces and exoskeletal motor torques. For the EAW (Bushing Forces) condition, joint moments from the EMG-tracked muscle driven simulations comprised contributions from muscle forces and interaction forces applied at the locations of the straps and the pelvic band. For the EAW (Prescribed Torques + Bushing Forces) condition, joint moments from the EMG-tracked muscle driven simulations comprised contributions from muscle forces, exoskeletal motor torques, and interaction forces applied at the locations of the straps and the pelvic band. These different simulation approaches provide new insights into the dynamics of the human-robot system, and systematically separate the contributions of the human and robot during EAW.

A limitation of this study is that the hip, knee, and ankle joint forces reported here are from an able-bodied participant, but the ReWalk P6.0 was designed for persons with SCI and other neurological conditions. The generalizability of our joint force results from an able-bodied participant to persons with SCI is limited because of substantially different musculoskeletal structure, motor control, and muscle tone. The goal of this study was to develop a novel computational framework to quantify the hip, knee, and ankle joint forces during EAW. Testing this computational framework on unassisted walking from an able-bodied participant permitted direct comparison of the joint force results with existing *in vivo* and simulation data. Furthermore, we simulated EAW using four different methods of increasing complexity to gain confidence in our computational framework. We are actively working on expanding this computational framework to quantify the hip, knee, and ankle joint forces in persons with SCI. The methodology to compute the intermediate model outputs, including joint kinematics and joint moments, are the same between AB and SCI participants. We have added a supplemental figure as pilot data from a participant with SCI during EAW (S5 Fig). A second limitation of this study is that external crutch reaction forces were not recorded during our EAW experiments. The handheld crutches permit additional stability and likely offload the lower limb joints during EAW. These crutch forces were not included in our virtual simulator, and hence, the effects of off-loading by the crutches on hip, knee, and ankle joint forces during EAW remain unclear. It is likely that the absence of crutch forces resulted in overestimation of joint loads during EAW. Furthermore, it is difficult to design a meaningful sensitivity analysis without quantifying which portions of EAW gait are affected by crutches, and the amount of potential offloading as a percentage of body weight. Prior literature estimates are limited to shoulder joint forces. Two prior studies have modeled crutch-assisted EAW with persons with SCI in a non-FDA-approved exoskeleton to quantify shoulder joint

forces [66,67]. Other experimental and modeling studies have quantified the effects of crutches on non-EAW gait [68–70]. Haubert and colleagues quantified shoulder joint forces during ambulation in persons with SCI using instrumented bilateral forearm crutches [68]. Segura and Piazza investigated the mechanics of ambulation with standard and spring-loaded crutches in able-bodied participants [69]. Perez-Rizo and colleagues developed a biomechanical model to study shoulder biomechanics in persons with SCI walking with crutches in two different gait patterns [70]. To the best of our knowledge, no prior study has estimated the effects of crutches on lower extremity joint forces. A third potential limitation is that there is no *in vivo* or experimental reference for joint loads during EAW, making it difficult to assess the validity of the computed forces [71]. As such, the absolute accuracy of joint load predictions during EAW remains uncertain.

This study provides a computational framework to quantify the hip, knee, and ankle joint forces during EAW in the ReWalk P6.0. Our approach provides a low-risk and cost-effective technique to quantify the loading of the long bones during EAW. This framework is applicable to other custom or FDA-approved exoskeletons, and lays the foundation for future studies to characterize the human-robot system to improve user safety and accelerate design refinements of wearable robotic exoskeletons. This computational framework can be incorporated into future EAW trials and clinical-use registries to better predict risk of fracture in persons with SCI depending on each user's bone and anthropometric variables, exoskeletal settings, and training parameters. Our methods have been described in adequate detail for the research community to reproduce our work. In addition, all empirical data used in this study will be available through an online repository. We invite other investigators to build on our work to develop virtual simulators of EAW and address a broad range of research questions.

## Supporting information

**S1 Fig. Left leg anterior-posterior (A-P) joint forces from the four approaches to simulate human-robot interactions during EAW.** Average (±1 SD) hip (A-D), knee (E-H), and ankle (I-L) joint forces during EAW (six trials, colored lines) were compared to unassisted walking (five trials, grey lines) and previously published *in vivo* (solid black lines) and simulated (dotted black lines) joint forces during unassisted walking. The joint forces were normalized to the participant's body weight (BW). Average toe-off from all unassisted walking and EAW trials is represented by dashed vertical lines. The braking and propulsion phases of gait during unassisted walking are represented by dotted vertical lines. (TIF)

**S2 Fig. Right leg anterior-posterior (A-P) joint forces from the four approaches to simulate human-robot interactions during EAW.** Average (±1 SD) hip (A-D), knee (E-H), and ankle (I-L) joint forces during EAW (six trials, colored lines) were compared to unassisted walking (five trials, grey lines) and previously published *in vivo* (solid black lines) and simulated (dotted black lines) joint forces during unassisted walking. The joint forces were normalized to the participant's body weight (BW). Average toe-off from all unassisted walking and EAW trials is represented by dashed vertical lines. The braking and propulsion phases of gait during unassisted walking are represented by dotted vertical lines. (TIF)

**S3 Fig. Left leg medial-lateral (M-L) joint forces from the four approaches to simulate human-robot interactions during EAW.** Average (±1 SD) hip (A-D), knee (E-H), and ankle (I-L) joint forces during EAW (six trials, colored lines) were compared to unassisted walking (five trials, grey lines) and previously published *in vivo* (solid black lines) and simulated (dotted black lines) joint forces during unassisted walking. The joint forces were normalized to the participant's body weight (BW). Average toe-off from all unassisted walking and EAW trials is represented by dashed vertical lines. The braking and propulsion phases of gait during unassisted walking are represented by dotted vertical lines. (TIF)

**S4 Fig. Right leg medial-lateral (M-L) joint forces from the four approaches to simulate human-robot interactions during EAW.** Average (±1 SD) hip (A-D), knee (E-H), and ankle (I-L) joint forces during EAW (six trials, colored

lines) were compared to unassisted walking (five trials, grey lines) and previously published *in vivo* (solid black lines) and simulated (dotted black lines) joint forces during unassisted walking. The joint forces were normalized to the participant's body weight (BW). Average toe-off from all unassisted walking and EAW trials is represented by dashed vertical lines. The braking and propulsion phases of gait during unassisted walking are represented by dotted vertical lines.
(TIF)

**S5 Fig. Average (±1 SD) hip, knee, and ankle joint angles (A-C), joint moments (D-F), and joint forces (G-I) from a participant with SCI (right leg: dark red lines, left leg: dark blue lines) and an able-bodied participant (right leg: light red lines, left leg: light blue lines).** The joint moments were normalized to the combined mass of each participant and the exoskeleton. The joint forces were normalized to each participant's body weight (BW). Average toe-off from all EAW trials of the participant with SCI is represented by dashed vertical lines.
(TIF)

## Acknowledgments

We thank Mr. William Kuo, Dr. Christopher Cirnigliaro, Mr. Michael Elliott, Ms. Ashley Pauly, Ms. Lynnette Chan, Mr. Denis Doyle Green, and Mr. David Kim for their assistance with EAW training and motion capture experiments. We thank Mr. Assaf Tzioni from ReWalk Robotics for his support with access to the exoskeleton encoder data. We thank Ms. Lina Alsauskaite from ReWalk Robotics for training the research team on safe use of the exoskeleton. ReWalk Robotics did not play any role in study design, participant recruitment, methodology, or data analysis.

## Author contributions

**Conceptualization:** Gabriela B. De Carvalho, Vishnu D. Chandran, William A. Bauman, Saikat Pal.

**Data curation:** Gabriela B. De Carvalho, Vishnu D. Chandran, Saikat Pal.

**Formal analysis:** Gabriela B. De Carvalho, Vishnu D. Chandran, Saikat Pal.

**Funding acquisition:** William A. Bauman, Saikat Pal.

**Investigation:** Vishnu D. Chandran, Saikat Pal.

**Methodology:** Gabriela B. De Carvalho, Vishnu D. Chandran, Saikat Pal.

**Project administration:** Ann M. Spungen, Noam Y. Harel, Saikat Pal.

**Resources:** Ann M. Spungen, Noam Y. Harel, William A. Bauman, Saikat Pal.

**Software:** Gabriela B. De Carvalho, Vishnu D. Chandran.

**Supervision:** Saikat Pal.

**Validation:** Gabriela B. De Carvalho, Vishnu D. Chandran, Saikat Pal.

**Visualization:** Gabriela B. De Carvalho, Vishnu D. Chandran, Saikat Pal.

**Writing – original draft:** Gabriela B. De Carvalho, Saikat Pal.

**Writing – review & editing:** Gabriela B. De Carvalho, Vishnu D. Chandran, Ann M. Spungen, Noam Y. Harel, William A. Bauman.

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
