## [Decision Letter · Decision Letter 0]

10 Jun 2025

Dear Dr. Pal,

Thank you for submitting your manuscript to PLOS ONE. After careful consideration, we feel that it has merit but does not fully meet PLOS ONE’s publication criteria as it currently stands. Therefore, we invite you to submit a revised version of the manuscript that addresses the points raised during the review process.

We look forward to receiving your revised manuscript.

Kind regards,

Jyotindra Narayan

Academic Editor

PLOS ONE

Journal Requirements:

4. Please note that PLOS ONE has specific guidelines on code sharing for submissions in which author-generated code underpins the findings in the manuscript. In these cases, we expect all author-generated code to be made available without restrictions upon publication of the work. Please review our guidelines at https://journals.plos.org/plosone/s/materials-and-software-sharing#loc-sharing-code and ensure that your code is shared in a way that follows best practice and facilitates reproducibility and reuse.

Additional Editor Comments:

The manuscript presents a computational framework to estimate hip, knee, and ankle joint forces during exoskeleton-assisted walking using the ReWalk P6.0 device, comparing four simulation approaches and validating results against in vivo and prior simulation data. Reviewers acknowledged the methodological rigor and writing quality but highlighted key concerns: limited generalizability due to reliance on a single able-bodied participant, exclusion of crutch forces, lack of ground truth for EAW validation, insufficient statistical analysis, and limited participants. Additional critiques emphasized missing demographic details, unclear method rationale, inadequate result interpretation, and the need for more relevant citations and figures to strengthen the manuscript’s impact and applicability.

Reviewers' comments:

Reviewer's Responses to Questions

**Comments to the Author**

1. Is the manuscript technically sound, and do the data support the conclusions?

Reviewer #1: Yes

Reviewer #2: Partly

Reviewer #3: Partly

Reviewer #4: Partly

2. Has the statistical analysis been performed appropriately and rigorously?

Reviewer #1: No

Reviewer #2: No

Reviewer #3: Yes

Reviewer #4: Yes

3. Have the authors made all data underlying the findings in their manuscript fully available?

Reviewer #1: Yes

Reviewer #2: No

Reviewer #3: Yes

Reviewer #4: Yes

4. Is the manuscript presented in an intelligible fashion and written in standard English?

Reviewer #1: Yes

Reviewer #2: Yes

Reviewer #3: Yes

Reviewer #4: Yes

Reviewer #1: This manuscript presents a computational framework for estimating hip, knee, and ankle joint forces during exoskeletal-assisted walking (EAW) using the ReWalk P6.0 device. The study compares four simulation approaches for modeling human-robot interaction and validates the framework by comparing simulated results with in vivo and prior simulation data. The manuscript is methodologically thorough and well-written, with detailed justification for modeling decisions and simulation parameters. However, despite its progress, significant concerns remain unaddressed. Some of the raised concerns are mentioedn below:

Major Concerns

1. Single-Participant Limitation

o While the authors justify the use of an able-bodied participant for validation, the generalizability of the findings is extremely limited. The target population (SCI patients) differs substantially in musculoskeletal structure, motor control, and muscle tone, and this should be better contextualized. A plan to extend the framework to SCI users is only briefly mentioned and should be strengthened or supplemented with pilot data.

2. Absence of Crutch Force Data

o The omission of external crutch forces in the simulation is a major limitation, as these are likely to offload lower limb joints significantly. Their exclusion could overestimate joint loading during EAW. This should be addressed explicitly in both the limitations and in the interpretation of results.

3. No Ground Truth for EAW

o There is no in vivo or experimental reference for joint loads during EAW, making it difficult to assess the validity of the computed forces. While the comparison to unassisted walking and other simulations is useful, the manuscript should make clearer that absolute accuracy of EAW predictions remains uncertain.

4. Statistical Analysis Lacking

o While the authors provide average and standard deviation values and RMS errors, there is a lack of formal statistical comparison (e.g., ANOVA or t-tests) between simulation conditions or against prior studies. Even though n=1, formal statistical testing across trials (where possible) would add rigor.

5. Figures and Interpretability

o Figures are numerous and detailed but may be overwhelming to readers. Consider consolidating or simplifying where appropriate. Moreover, the use of color coding across simulation methods should be standardized across all plots to aid interpretation.

Minor Concerns

• Terminology: The term “functionally equivalent to physical EAW” may be too strong when describing the “Prescribed Torques + Bushing Forces” method. Consider tempering this language.

• Rationale for EMG Muscles: A clearer rationale for selecting the eight muscles used for EMG tracking would improve transparency.

• Clarify Impact of Torques vs. Bushing Forces: While the distinction between methods is clear, a clearer discussion of their individual biomechanical impact on each joint would improve mechanistic interpretation.

• Literature Citations: Some references, such as those for previously published in vivo data, are briefly mentioned but not thoroughly discussed. More critical evaluation of how current findings extend or deviate from prior work is needed.

Recommendations for Improvement

1. Consider conducting simulations on data from at least one SCI participant, even if exploratory, to illustrate feasibility and differences.

2. Quantify the potential error introduced by the exclusion of crutch forces—either via sensitivity analysis or literature estimates.

3. Improve clarity in figures, possibly through simplification or use of summary plots comparing peak forces across methods.

4. Include formal statistical tests, even within-subject, where appropriate.

5. References: The following refeercnes are highly recommended toa dded to the paper to improve its quality and state of the arts. (10.1007/s41315-025-00421-x), (10.22024/UniKent/01.02.105160), and (10.1109/ACCESS.2023.3325211).

Reviewer #2: The paper presents the comparative study on Hip, knee, and ankle joint forces during exoskeleton-assisted walking. The work is interesting. The following are the suggestions to incorporate:

1. The novelty and outcome of the work are to be re-phrased.

2. The future use of the this work should be clearly explained in order to support the applicability of the outcome of this work.

3. In the Introduction, more recent relevant work should be cited related to problem formulation.

4. Caption of Fig 1 (A) and (B) is not clear.

5. The authors have not discussed about the participants demographic information such as age, height.

6. How many trials of walking have been used for this study?

7. Did the virtual simulator generated use the same anthropomorphic dimension as that of each participants?

8. The presentation of study is not good. The multiple things cause confusion to the reader. Please manage the subsections of methods section appropriately.

9. The description of Results is not sufficient and can be improved further.

10. Why did authors choose those four specific methods for EAW.

11. References are not justified

12. Discussion can be re-phrased in order to enhance clarity of the study.

13. The study should be extended for experiments with SCI participants.

Reviewer #3: The manuscript entitled ‘’Hip, knee, and ankle joint forces during exoskeletal-assisted walking: comparison of approaches to simulate human-robot interactions” has been organized and developed in good shape. The overall goal of this study was to develop a computational framework to quantify hip, knee, and ankle joint forces during exoskeletal-assisted walking (EAW) in the ReWalk P6.0, an FDA approved lower-extremity exoskeleton. It has 2 main objectives: The first objective was to quantify hip, knee, and ankle joint forces during unassisted walking and compare the results to existing in vivo and simulation data. The second objective was to compare hip, knee, and ankle joint forces from four different approaches to simulate human-robot interactions during EAW. OpenSim Moco is used to determine the joint reaction forces at the hips, knees, and ankles during unassisted walking and EAW. The study is well-developed and the results are intriguing. Once the following comments are addressed, the manuscript is recommended for publication. Please go through the attached document.

Reviewer #4: This research article focuses on developing a computational framework to quantify the hip, knee, and ankle joint forces during exoskeleton-assisted walking. Although the manuscript is well-written, I have the following concerns, which may help the authors to improve the article.

1) The motivation behind this research is not clearly highlighted in the manuscript. There exist standard protocols as well as real-time approaches to evaluate joint torque during walking. This manuscript did not adequately justify how this work contributes in comparison to the existing literature that quantified joint moments.

2) How different and novel is this framework compared to the existing studies that can evaluate the joint moments during assisted walking?

3) The framework used for custom-made exoskeletons can be utilized to measure the joint moments for other types of exoskeletons. What difference will the proposed framework make?

4) The study was validated with one subject, which raises questions about whether this framework works for a different subject or a variety of subjects.

5) The methodology needs to be explained more clearly with figures of the experiment with the real subject.

6) What conclusions can be drawn from Fig. 2? Is it only the results or a comparison with the earlier studies?

**Do you want your identity to be public for this peer review?** For information about this choice, including consent withdrawal, please see our Privacy Policy

Reviewer #1: No

Reviewer #2: No

Reviewer #3: **Yes: ** NEHA SAHU

Reviewer #4: No

---

## [Author Response · Author response to Decision Letter 1]

14 Jul 2025

Please see the attached document.

---

## [Decision Letter · Decision Letter 1]

8 Aug 2025

Hip, knee, and ankle joint forces during exoskeletal-assisted walking: comparison of approaches to simulate human-robot interactions

PONE-D-25-14353R1

Dear Dr. Pal,

We’re pleased to inform you that your manuscript has been judged scientifically suitable for publication and will be formally accepted for publication once it meets all outstanding technical requirements.

Kind regards,

Jyotindra Narayan

Academic Editor

PLOS ONE

Additional Editor Comments (optional):

The reviewers have now accepted the revised work for publication. Therefore, I recommend the same for publication. Congratulations to the authors. 

Reviewers' comments:

Reviewer's Responses to Questions

**Comments to the Author**

Reviewer #1: All comments have been addressed

Reviewer #3: All comments have been addressed

Reviewer #4: All comments have been addressed

2. Is the manuscript technically sound, and do the data support the conclusions?

Reviewer #1: Yes

Reviewer #3: Yes

Reviewer #4: Yes

3. Has the statistical analysis been performed appropriately and rigorously?

Reviewer #1: Yes

Reviewer #3: Yes

Reviewer #4: Yes

4. Have the authors made all data underlying the findings in their manuscript fully available?

Reviewer #1: Yes

Reviewer #3: Yes

Reviewer #4: Yes

5. Is the manuscript presented in an intelligible fashion and written in standard English?

Reviewer #1: Yes

Reviewer #3: Yes

Reviewer #4: Yes

Reviewer #1: The revised version of the paper has been improved significantly and is highly recommended for publication.

Reviewer #3: The manuscript entitled “Hip, knee, and ankle joint forces during exoskeletal-assisted walking:

comparison of approaches to simulate human-robot interactions” has been organized and

developed in good shape. The overall goal of this study was to develop a computational framework

to quantify hip, knee, and ankle joint forces during exoskeletal-assisted walking (EAW) in the

ReWalk P6.0, an FDA approved lower-extremity exoskeleton. It has 2 main objectives: The first

objective was to quantify hip, knee, and ankle joint forces during unassisted walking and compare

the results to existing in vivo and simulation data. The second objective was to compare hip, knee,

and ankle joint forces from four different approaches to simulate human-robot interactions during

EAW. OpenSim MOCO is used to determine the joint reaction forces at the hips, knees, and ankles

during unassisted walking and EAW. The study is well-developed and the results are intriguing.

Once the following comments are addressed, the manuscript is recommended for publication.

Reviewer #4: The authors have adequately addressed my comments and modified the manuscript accordingly. I have no further comments.

**Do you want your identity to be public for this peer review?** For information about this choice, including consent withdrawal, please see our Privacy Policy

Reviewer #1: No

Reviewer #3: **Yes: ** Neha Sahu

Reviewer #4: No

---

## [Editor Report · Acceptance letter]

PONE-D-25-14353R1

PLOS ONE

Dear Dr. Pal,

I'm pleased to inform you that your manuscript has been deemed suitable for publication in PLOS ONE. Congratulations! Your manuscript is now being handed over to our production team.

Kind regards,

on behalf of

Dr. Jyotindra Narayan

Academic Editor

PLOS ONE